# Invasive Bacterial Infections of the Musculoskeletal and Central Nervous System during Pig Rearing: Detection Frequencies of Different Pathogens and Specific *Streptococcus suis* Genotypes

**DOI:** 10.3390/vetsci11010017

**Published:** 2024-01-02

**Authors:** Ninette Natascha Bornemann, Leonie Mayer, Sonia Lacouture, Marcelo Gottschalk, Christoph Georg Baums, Katrin Strutzberg-Minder

**Affiliations:** 1IVD Innovative Veterinary Diagnostics (IVD GmbH), Albert-Einstein-Str. 5, 30926 Seelze, Germany; 2Institute of Bacteriology and Mycology, Centre for Infectious Diseases, Faculty of Veterinary Medicine, University of Leipzig, 04103 Leipzig, Germanychristoph.baums@vetmed.uni-leipzig.de (C.G.B.); 3Faculty of Veterinary Medicine, University of Montreal, Saint-Hyacinthe, QC J2S 2M2, Canada; sonia.lacouture@umontreal.ca (S.L.); marcelo.gottschalk@umontreal.ca (M.G.)

**Keywords:** arthritis, meningitis, *Streptococcus suis*, type of capsule, lymph node

## Abstract

**Simple Summary:**

Diseases of the locomotor system and the central nervous system (CNS) often lead to herd problems during pig rearing. Bacterial infections are the most common cause. This study aimed to clarify which pathogens are detectable in these diseases in Germany and whether *Streptococcus suis* (*S. suis)* can be detected in the lymph nodes of naturally infected pigs. In 201 pigs with a clinical history of joint disease, meningitis or sudden death, joint and meningeal swabs, kidney, lung, and eight different lymph nodes were examined culturally. Isolated pathogens were identified using a polymerase chain reaction. The capsule typing of *S. suis* was carried out using a multiplex polymerase chain reaction, including the detection of three genes to differentiate the pathotype. The results of the present study are limited by the selected study population as no representative sample was examined. Nonetheless, the results highlight the importance of *S. suis* as a causative agent of arthritis and meningitis in pigs, especially in weaning piglets. There are age-specific differences concerning the frequency of the detectable *cps* types. The systematic examination of various lymph nodes indicates that the host–pathogen interaction of *S. suis* in lymph nodes is an important part of systemic infection.

**Abstract:**

Locomotor and central nervous system disorders occur during pig rearing, but there is no systematic recording of the different causative agents in Germany. Joint and meningeal swabs, kidneys, lungs, and eight different lymph nodes per pig were cultured, and isolated pathogens were identified using polymerase chain reactions (PCRs). The *cps* and pathotype of *Streptococcus suis* (*S. suis*) isolates were determined using multiplex-PCR. *S. suis* was the most important pathogen in the infected joints (70.8%) and meningeal swabs (85.4%) and was most frequently detected in both sites in suckling and weaning piglets. To elucidate the possible portal of entry of *S. suis*, eight different lymph nodes from 201 pigs were examined in a prospective study. *S. suis* was detected in all examined lymph nodes (*n* = 1569), including the mesenteric lymph nodes (15.8%; *n* = 121/765), with *cps* 9 (37.2%; *n* = 147) and *cps* 2 (24.3%; *n* = 96) being the most dominating *cps* types. In piglets with a systemic *S. suis* infection, different lymph nodes are frequently infected with the invasive *S. suis* strain, which does not help clarify the portal of entry for *S. suis*.

## 1. Introduction

Diseases of the locomotor and central nervous systems affect the economic aspects of both pig production and animal welfare [1]. The possible aetiologies of clinical and subclinical musculoskeletal and central nervous disorders are numerous [1,2], but there is, in general, little information on the frequency distribution of the causes.

*Streptococcus suis* (*S. suis*) causes polyarthritis and meningitis as well as septicaemia, endocarditis, and sudden death [3]. At the same time, other bacterial pathogens can also lead to arthritis and meningitis in pigs, such as *Escherichia coli* (*E. coli*), *Glaesserella parasuis* (*G. parasuis*), and *Mycoplasma* (*M*.) *hyorhinis* [1,2]. Other diseases, such as enterotoxaemia or salt poisoning, may also lead to clinical signs of affected central nervous systems [2].

Although *S. suis* isolates are genotyped on a large scale in various European countries, there are no current studies on the differences in the prevalence of *S. suis* genotypes between different age classes of pigs. However, this is of great importance for herd management and the use of herd-specific autogenous vaccines [4]. Pruefer et al., (2019) recently differentiated serotypes in a larger collection of invasive *S. suis* isolates collected in Germany [5]. The authors classify the *S. suis* isolates based on the original tissue of isolation. As these are isolates from internal organs or swabs, often no further information is available, e.g., the age class of the affected pig and the pathological findings of the sampled site [5].

*S. suis* infections can cause diseases in pigs of all ages [6]. Weaning piglets, in particular, are susceptible to *S. suis* disease [7]. Wisselink et al., (2000) detected *S. suis* serotype 1 particularly frequently in suckling piglets [8]. Diseases caused by serotype 7 are mainly to be expected at the age of six weeks, whereas serotype 2 occurs very frequently in animals between three and twelve weeks of age or older [9,10,11,12]. However, there is a lack of studies on the age class dependency of the other *cps* types.

Although *S. suis* is one of the most important pathogens of pigs worldwide, the exact mechanisms leading to the colonisation of the host are still widely unknown [13]. Infections of pigs probably occur mainly via the upper respiratory tract [3,14,15]. In contrast, injured skin sites are an important portal of entry for human infections with *S. suis* [14,16]. However, in recent years, the gastrointestinal route of infection in humans has become the focus of attention, especially in South Asian countries [14,15,17]. As a result, the possibility of gastrointestinal infection has also been highlighted in various infection trials in pigs [18,19,20]. In this context, the gastrointestinal lymph nodes of the pigs were often examined to prove that the pathogen was translocated via the gastrointestinal tract. Evidence of gastrointestinal infection was found bypassing the stomach [18,19], while in an infection feeding trial, *S. suis* could not be detected in the gastrointestinal lymph nodes [20].

In the present study, we investigated the frequency distribution of bacterial causes of arthritis and meningitis in Germany by combining necropsy, culture, and histopathology to update the information on the role of various pathogens in the field. Pathogen distribution was examined by age class, and *S. suis* isolates (the most common isolated pathogen) were genotypically differentiated. In addition, various lymph nodes were culturally investigated to elucidate the putative portal of entry of natural *S. suis* infections in pigs.

## 2. Materials and Methods

### 2.1. Animals

Animals were selected from submissions to the Innovative Veterinary Diagnostic Laboratory (IVD GmbH), Seelze, Germany, for diagnostic purposes, independent from this study, during a 15-month period, from May 2016 to July 2017. A pig was only included if the complete carcass was available for necropsy, and (1) the carcass showed macroscopical lesions, indicating either arthritis or exudative meningitis, (2) the preliminary report included clinical signs of central nervous disorder (incoordination, recumbency with paddling), or (3) the preliminary report was sudden death, and the carcass was free of lesions indicating any other specific disease/cause of death. A total of 201 pigs were included in this study.

This study was not an animal experiment. The pigs were delivered to IVD GmbH (a commercial diagnostic laboratory) for diagnostic purposes only. Animal owners submitted their pigs for necropsy, and further diagnostics were absolutely independent from this study. The euthanasia of the pigs was necessary because they were seriously diseased, and further treatment would no longer have been possible. The animals were not killed in the herd if there was no vet on site or the animal owner could not carry out the killing himself in accordance with animal welfare standards. The fact that this study was carried out had no influence on the decision to euthanise. Euthanasia was performed only for diagnostic and not for scientific reasons.

### 2.2. Background Data and Gross Examination

For each pig, the age category of the animal (suckling piglet, weaning piglet, fattening pig), the pre-treatment, and the identity of the herd of origin were recorded. While the weight of suckling and weaning piglets was estimated, fattening pigs were weighed. Within four hours of euthanasia or within 24 h of natural death, all pigs were subjected to a complete necropsy, and all gross lesions were recorded.

### 2.3. Tissue Selection

Representative tissues and/or swabs for bacteriology and histopathology were selected from each necropsied pig. Swab samples were taken from the affected joint and/or CNS. These included swabs placed in Amies medium (Nerbe plus GmbH & Co. KG, Winsen/Luhe, Germany) and nylon flock fibre swabs without a medium (Hain Lifescience GmbH, Nehren, Germany).

Tissue samples for bacteriological and histopathological investigations were taken from the *Lobus medius* of the lung, the caudal part of the right kidney, and seven lymph nodes: *Lymphonodus* (*Ln*.) *cervicalis superficialis dorsalis dexter*, *Ln. bifurcationis medii*, *Ln. gastricus*, *Ln. jejunalis*, *Ln. ileocolicus*, *Ln. iliaci medialis dexter*, and *Ln. inguinalis superficialis dexter.* From the 40th pig onwards, the *Ln. colicus* was also sampled to include a lymph node of the large intestine in the examination. Other sites such as the serosa, endocard (“heart valve”), spleen, and bursa were sampled in case of gross pathological visible lesions. Tissue samples for histopathological examinations were fixed in 10% formaldehyde.

### 2.4. Bacteriology

The cultural examination of swabs and organ samples was performed immediately after necropsy. Cultural examinations of lymph nodes were performed within eight hours after necropsy. In a few cases, lymph nodes were refrigerated overnight (4 °C) and culturally examined the next day.

The tissue samples were fractionated on Columbia blood agar and streptococcal staphylococcal selective agar (Oxoid, Wesel, Germany). Swab samples from the joint, meninges, bursa, and serosa were also fractionated on Columbia blood agar, streptococcal staphylococcal selective agar (Oxoid, Wesel, Germany), chocolate agar with addition of nicotinamide adenine dinucleotide (NAD) (in-house production) and Gassner agar as a selective culture medium for differentiation of lactose positive and negative *Enterobacteriaceae* (Oxoid, Wesel, Germany). The joint, bursa, and serosa swabs were also streaked onto Mycoplasma Experience solid (MEX) culture medium (Mycoplasma Experience Ltd., Surrey, Great Britain). On all blood culture media inoculated with lungs, joint, bursa, serosa or meningeal swabs, an inoculation loop with *S. epidermidis* as NAD nurse was added to detect NAD-dependent bacteria. Cultures were incubated at 37 °C with 5% CO_2_ on agar, and growth was examined at 24 and 48 h, except MEX medium incubated at 37 °C with 5% CO_2_, and growth was examined at 72 and 96 h. Isolates were identified by standard laboratory procedures based on morphological characteristics and molecular biology (specific PCRs). *S. suis* isolates were stored at −80 °C in 1 mL THB with 20% glycerine.

Bacteriologically culture-negative joint swabs and meningeal swabs were tested by PCR using an in-house multiplex PCR (MP PCR) to detect the presence of *G. parasuis*, *M. hyorhinis*, *M. hyosynoviae*, and *S. suis*.

### 2.5. cps Typing and Virulence-Associated Gene Profiling

Screening of the *S. suis* isolates by MP PCR, which identified *cps* types 1 or 14, 2 or 1/2, 7, and 9, was conducted essentially as described [21], but with different primers for *cps* 1, *epf* and *gdh*, as specified in Table 1. Also, primers for *srtD* were added, while primers for *arcA* were not used. Due to primer modifications, no variants of the *epf* gene could be differentiated.

MP PCR was performed using lysates of *S. suis* colonies grown on blood agar plates. Bacterial suspensions were diluted in 50 μL lysis buffer and heated for 60 min at 60 °C in an incubator and 15 min at 97 °C in a heating block. Ultimately, 1.5 μL of the lysate was used as a template.

All *S. suis* isolates that did not belong to one of the examined *cps* types in the previously performed MP PCR to detect *cps* types 1 or 14, 2 or 1/2, 7, and 9, were examined by PCR for *S. suis cps* 4 using primer sequences as published before [23].

Further *cps* typing and distinguishing of *cps* types 1 and 14, as well as *cps* types 2 and 1/2, were carried out as follows: A representative number of *S. suis* isolates were prepared as lyophilizates and sent to the University of Montreal, St-Hyacinthe, QC, Canada, for complete serotyping (all 35 originally described serotypes) and differentiation of serotypes 2-1/2 and 1-14. Indeed, 36 isolates sent were not *cps* type 1 or 14, 2 or 1/2, 4, 7, or 9. MP PCR tests for the determination of all known *cps* types were carried out on these isolates [24]. A mismatch amplification mutation assay (MAMA) to distinguish *cps* types 2 and 1/2 was performed from 33 *cps* 2 or 1/2 isolates [25]. For the differentiation of *cps* 1 and 14, the *cps* K gene of 15 *cps* 1J positive isolates was conducted as described [26].

### 2.6. Histopathology

All synovial membranes and brains were subjected to histopathological examination. After fixation in 10% neutral buffered formaldehyde, the tissues were processed through graded concentrations of ethanol and xylene and embedded in paraffin wax. Tissue sections were stained with haematoxylin and eosin (HE).

### 2.7. Statistical Analysis

An Excel file was used to store information obtained from each case concerning the epidemiologic factors, preliminary reports, and gross and microscopic lesions. Collected data were used for descriptive statistics. Chi-square test and Fisher exact test were performed. Probabilities lower than 0.05 were considered significant.

## 3. Results

### 3.1. Gross Lesions and Background Data

In total, 201 pigs (27 suckling piglets, 144 weaning piglets and 30 fattening pigs) from 95 farms were sampled. An examination of piglets was generally only initiated when the animals showed clinical signs of severe acute disease typical of a herd problem. The weight of the pigs ranged from 2.5 kg to 79 kg. The number of animals from each herd ranged from one to twelve. The most common necropsy findings were polyarthritis (*n* = 88), arthritis (*n* = 31), opacity and hyperaemia of the pia mater (*n* = 77), and multiple oedema (*n* = 36). Fibrinous and/or suppurative pleuritis, epicarditis, pericarditis, peritonitis, polyserositis, and endocarditis were also reported in a few cases.

### 3.2. Histopathology

Generally, the histopathological examination confirmed the necropsy diagnoses and revealed exudative (i.e., suppurative or fibrinopurulent) lesions in the brain and joints, as shown in Figure 1 and Figure 2. A total of 117 joint capsules were pathohistologically examined. Synovitis was diagnosed in all 117 tissue samples, with 112 cases also showing luminal inflammatory cell infiltrates, thereby confirming an arthritis diagnosis. The synovitis was acute in 94 tissue samples (80.3%), and it was classified as severe in 69 cases (59.0%). In most joint capsules studied, the arthritis was diffuse (*n* = 113). The character of the inflammation was fibrinopurulent in most cases (94.0%; *n* = 110).

The tissue samples of the CNS were also pathohistologically examined from 133 animals. In 27 animals (20.3%), pathohistological lesions of meningoencephalitis were observed. In 61 cases (45.9%), leptomeningitis was diagnosed. In one case (0.8%), there were both lesions indicative of leptomeningitis and lesions indicative of cerebrospinal angiopathy. Lesions indicative of cerebrospinal angiopathy were found pathohistologically in 28 animals (21.1%) only.

Pathohistological signs of meningoencephalitis or leptomeningitis were found in 88 (66.2%) of 133 samples, and they were in the majority fibrinopurulent (*n* = 64) category. In 77 pigs, an inflammation of the meninges had been suspected based on gross lesions, which, in turn, was confirmed pathohistologically in 75 cases. This underlines the importance of the fine tissue examination for a more exact diagnosis in the presence of central nervous symptoms.

### 3.3. Bacteriology of Joint and Meningeal Swabs

To identify the infectious causes of arthritis, 120 joint swabs from 119 animals with gross inflammatory joint lesions were cultured. A single isolate was isolated from 76 joints while two different pathogens were recovered in two affected joints (Table 2). *S. suis* was cultured in 46 of 120 joint swabs (38.3%), and bacteria other than *S. suis* were detected in 31 joint swabs (25.8%).

Seventy-seven pathogens were cultured from 76 meningeal swabs from 133 animals (Table 3). Two pathogens (*S. suis* and *Staphylococcus aureus* (*S. aureus*), compared in Table 3), were detectable in one meningeal swab. *S. suis* was present in 48.1% of the meningeal swabs (*n* = 64). *G. parasuis* and *Trueperella pyogenes* (*T. pyogenes*) were isolated from three meningeal swabs each (2.2%).

### 3.4. Arthritis: Association between Detected Pathogens, Age Category, and Histopathological Lesions

A separate examination of the individual age classes was aimed at investigating whether there are age-dependent differences in the aetiology. The 120 joint swabs examined were from 24 suckling piglets (20.8%), 77 weaning piglets (64.2%), and 18 fattening pigs (15.0%). While there were no striking differences in the histopathological lesions of the joint capsules between the different age categories, differences were found in pathogen detection (Table 2).

In the bacteriological cultural examination of 25 joint swabs from 24 suckling piglets, *S. suis* was the most frequently detected pathogen (*n* = 6), followed by *E. coli* (*n* = 5). In half of the examined joint swabs from weaning piglets, *S. suis* was cultured (*n* = 39; 50.6%). Other pathogens, such as *M. hyorhinis* (*n* = 5) and *G. parasuis* (*n* = 3), were detected much less frequently. While the proportion of culturally negative joint swabs in suckling (*n* = 7) and weaning (*n* = 23) piglets was comparable (28.0% and 29.9%, respectively), the proportion in fattening pigs was almost three times greater (77.8%). Although arthritis (*n* = 16) or synovitis (*n* = 2) was pathohistologically detected in fattening pigs, pathogens were isolated in only four joint swabs (Table 2). Pathogens were significantly less frequently cultured in joint swabs from fattening pigs than suckling piglets (*p* = 0.002) and significantly less frequently than weaning piglets (*p* ≤ 0.001).

*S. suis* was cultured from joint swabs in 72.2% (*n* = 39) of the weaning piglets and thus significantly more frequently than in the suckling piglets (*p* = 0.036) with a proportion of 33.3% (*n* = 6) and significantly more frequently than in the fattening pigs (*p* ≤ 0.001) with 25% (*n* = 1). In contrast, *Staphylococcus hyicus* (*S. hyicus*) was significantly more frequently detectable in joint swabs of suckling piglets than in weaning piglets (*p* = 0.016). While *E. coli* was significantly more frequently detectable in suckling piglets than in weaning piglets (*p* ≤ 0.001), the difference with joint swabs from fattening pigs was not significant (*p* = 0.064).

A total of 44 negative joint swabs (suckling piglet: *n* = 7; weaning piglet: *n* = 23; fattening pig: *n* = 14) were examined by MP PCR to identify *G. parasuis*, *M. hyorhinis*, *M. hyosynoviae* and *S. suis*. Within nine joint swabs genome fragments of *M. hyorhinis* (*n* = 1), *M. hyosynoviae* (*n* = 5), *S. suis* (*n* = 2), and *S. suis* and *M. hyorhinis* (*n* = 1) were detected. In 35 of the 44 culturally negative joint swabs, the MP PCR results also remained negative. No genome fragments of the above pathogens were detectable in any of the suckling piglet swabs (100%), in 19 joint swabs of weaning piglets (82.6%), and in nine fattening pigs (64.3%). However, MP PCR was useful for detecting arthritis caused by *M. hyosynoviae*, which was detected in six joint swabs from fattening pigs; indeed, *M. hyosynoviae* was detected more frequently in joint swabs from fattening pigs than suckling (*p* = 0.003) and weaning piglets (*p* ≤ 0.001).

*S. suis* was most frequently detected in arthritis in both suckling and weaning piglets. However, other pathogens were also isolated in the joints of the same magnitude in suckling piglets, while bacterial pathogens other than *S. suis* were rarely present in weaning piglets.

### 3.5. CNS: Association between Detected Pathogens, Age Category, and Histopathological Lesions

The 133 examined meningeal swabs were from 7 suckling piglets (5.3%), 109 weaning piglets (81.9%) and 17 fattening pigs (12.8%). The number of examined weaning piglets was much larger. The reason was the high occurrence of clinical signs of central nervous dysfunction and sudden deaths in weaning piglets, which led to the suspected diagnosis of bacterial meningitis with a collection of meningeal swabs. Pathohistological lesions were visible in 130 CNS tissue samples. The frequency of cultural detection of the different pathogens in association with a pathohistological lesion and age category is shown in detail in Table 3.

In the seven CNS samples of suckling piglets, leptomeningitis (*n* = 6) or meningoencephalitis (*n* = 1) were observed. In five meningeal swabs, the following pathogens were isolated: *E. coli* (*n* = 1), *S. suis* (*n* = 2) and *T. pyogenes* (*n* = 2). In the two culturally negative meningeal swabs, *S. suis* (leptomeningitis) and *G. parasuis* (meningoencephalitis) were detectable by PCR only.

Leptomeningitis (*n* = 54) or meningoencephalitis (*n* = 19) were the most frequently observed lesions in CNS samples of weaning piglets (Table 3). *S. suis* was the only organism isolated from 78.1% (*n* = 57) of the cases in which leptomeningitis and meningoencephalitis were observed. In the remaining 21.9% (*n* = 16) *G. parasuis* (*n* = 3), *S. aureus* (*n* = 2), *T. pyogenes* (*n* = 1), or no pathogen (*n* = 10) could be isolated. Cerebrospinal angiopathy was observed in 21.1% of the weaning piglets (*n* = 23). In 17.4% (*n* = 4) of weaning piglets with cerebrospinal angiopathy, a pathogen could also be isolated.

In the pathohistological examination of the CNS samples of the fattening pigs (*n* = 17), hyperaemia (*n* = 4), cerebrospinal angiopathy (*n* = 5), fibrinous purulent leptomeningitis (*n* = 1), or meningoencephalitis (*n* = 7) were observed. Culturally, a pathogen was only detectable in three meningeal swabs from fattening pigs with meningoencephalitis. *S. suis* (12.5%) was culturally isolated in two meningeal swabs and *S. dysgalactiae* in one. No pathogen could be culturally detected in the remaining 14 meningeal swabs; no further pathogens could be confirmed by MP PCR as well.

Leptomeningitis was the most frequent lesion of the CNS in all pigs. At the same time, it occurred significantly less frequently in the fattening pigs than in the suckling piglets (*p* ≤ 0.001) and weaning piglets (*p* = 0.002). There were no significant differences between the age classes for the other CNS lesions. Although *S. suis* was detected in leptomeningitis and meningoencephalitis in all age categories, *S. suis*-associated leptomeningitis and meningoencephalitis were most frequently found in weaning piglets.

### 3.6. S. suis cps Types Isolated

In total, 529 *S. suis* isolates were recovered. *S. suis* was isolated in 46 of the 120 (38.3%) joint swabs and in 64 of the 133 (48.1%) meningeal swabs. *S. suis* was cultured in 51 of the 201 (25.4%) lungs and in 61 (30.3%) of the 201 kidneys. Of the 1569 lymph nodes examined, 270 *S. suis* isolates were detected in 268 lymph nodes (17.1%).

To investigate the distribution of the *cps* types in further detail, selected isolates were additionally typed by MP PCR, MAMA, and *cps* K sequencing. Each different genotype (which also were morphologically different) was tested at least once per herd. As not all *S. suis* isolates were typed and distinguished between *cps* 1 or 14 and *cps* 2 or *cps* 1/2, the isolates with the same morphology and genotype from the same herd and submission were assumed to be the identical *cps* type. Twenty-six *S. suis* isolates were tested by MP PCR. The isolation sites and results are summarized in Appendix A. By MP PCR, the *cps* type could be determined in 16 isolates and ten isolates could not be typed (Appendix A). Of 24 isolates from the lymph node that were further examined, six isolates remained non-typeable, and eight isolates were distributed over seven *cps* types (Appendix A). To distinguish between *cps* types 2 and 1/2, a MAMA was carried out on 23 *S. suis* isolates [25]. Based on this (Appendix A), fifteen isolates belonged to *cps* 2, and seven isolates to *cps* 1/2. One isolate could not be differentiated. Of the fifteen isolates that were isolated from meninges, twelve were determined as *cps* 2. This leads towards the assumption that *cps* 2 is more associated with central nervous Infections than *cps* 1/2. To distinguish *cps* 1 and *cps* 14, 15 isolates were analysed by *cps* K sequencing [26]. Based on the *cps* K sequence, thirteen isolates belonged to *cps* 1, and two isolates were *cps* 14, indicating that *cps* 1 is more common than *cps* 14 (as already reported by Mayer et al., 2021 [27]).

The results of the PCR-based capsule and pathotyping (muramidase-released protein gene *(mrp)*, suilysin gene *(sly)*, extracellular factor gene *(epf)*) of the isolates are shown in Appendix A as absolute frequencies of each *cps* type tested. The observed frequencies of virulence marker genes show that *mrp* and *sly* were relatively widespread, whereas *epf* was detected less frequently. The gene *epf* only occurred in combination with *sly* and was most prevalent in *cps* 14, 1, and 2. The *epf* gene was also present in *S. suis* isolates with *cps* types 1/2, 8, and 9 and in non-typable *S. suis* isolates with a relative proportion of 68.0% (*n* = 34), 10.0% (*n* = 1), 3.2% (*n* = 7), and 14.3% (*n* = 3), respectively.

### 3.7. S. suis Positive Animals

Out of the 201 animals tested, 124 animals (16 suckling piglets, 93 weaning piglets and 15 fattening pigs) were culturally *S. suis* positive in at least one sample (Table 4).

Out of the 529 isolates, 35 originated from suckling piglets, 463 from weaning piglets, and 31 from fattening pigs. For *S. suis* isolates from weaning piglets, the largest proportions were *cps* type 9 (41.7%, *n* = 193) and 2 (33.5%, *n* = 155). For suckling piglets, the most isolates were *cps* type 7 (48.6%, *n* = 17) and 1 (20.0%, *n* = 7) (Appendix A).

Of 124 *S. suis*-positive animals, only one site was positive for *S. suis* in 43 animals, and two or more sites were positive in 81 animals. 107 animals had only one or more isolates of the same genotype. In 17 animals, two different *S. suis* genotypes were detectable and in one animal four different genotypes. This involved the detection of one genotype in multiple sites and the detection of the other genotype in a single site. So, in most animals, *S. suis* could be isolated at more than one site (65.3%), and only one genotype could be detected (85.5%).

In 124 animals, *S. suis* could not be isolated in any lymph node. In 77 animals, *S. suis* was detectable in at least one lymph node, and in 52 animals, in more than one lymph node and in different sites. The detection of *S. suis* in multiple lymph nodes was associated with detection in multiple internal organs. Among the 25 animals with only one positive lymph node, there were 21 animals in which *S. suis* could be found in more than one site. If more than one lymph node is positive, it is not possible to determine how *S. suis* entered the body. In two animals with only one *S. suis*-positive lymph node, it was an *S. suis*-positive mesenteric lymph node.

The detection rate of *S. suis* in the different lymph nodes was comparable to some extent (Table 5). There was no indication of how *S. suis* enters the body, but this suggests the involvement of the lymph nodes in systemic *S. suis* infections.

A total of 155 *S. suis cps* 2 isolates were detected in 28 animals. The *cps* 2 *mrp*+ *sly*+ *epf*+ genotype was dominant with 134 (86.5%) *S. suis* isolates from 24 animals. This genotype was cultured in more than three sites in 15 weaning piglets. While *cps* 2 *mrp*+ *sly*+ *epf*+ was not detected in any lymph node in six weaning piglets, two or more lymph nodes and different other sites were positive for *cps* 2 *mrp*+ *sly*+ *epf*+ in 15 weaning piglets.

Two hundred and three *cps* 9 isolates were detected in 43 animals. The genotype *mrp*+ *sly*+ *epf*− was the most frequent (*n* = 181; 89.2%) and was detectable in 32 animals. In 21 pigs, *cps* 9 *mrp*+ *sly*+ *epf*− was detectable in more than three sites. In six animals, no *cps* 9 *mrp*+ *sly*+ *epf*− was detectable in any lymph node; in 18 animals, two or more lymph nodes and different other sites were *cps* 9 *mrp*+ *sly*+ *epf*− positive. The occurrence of other *cps* types like *cps* 1, 4, and 7 was mostly limited to a single or few sites. Therefore, it can be speculated that systemic infections occur more frequently with *cps* 2 and 9.

In summary, different lymph nodes of a severely diseased piglet are frequently infected with the same invasive *S. suis* strain in accordance with systemic dissemination.

### 3.8. Occurrence of S. suis Genotypes per Site

To avoid the over-representation of herds with multiple submissions, each different genotype (*cps*, *mrp*, *sly*, *epf*) was considered only once from each herd and each site. This resulted in a reduction to 395 isolates (Appendix A) with *cps* 9 being the most frequent one (37.2%; *n* = 147), followed by *cps* 2 (24.3%; *n* = 96). *Cps* 1, 1/2, 4 and 7 occurred less frequently and were detected in 5.1% (*n* = 20), 7.1% (*n* = 28), 7.1% (*n* = 28) and 6.8% (*n* = 27) of the isolates, respectively.

In the lymph nodes (*n* = 213), *cps* 9 occurred most frequently, with a relative proportion of 38.5% (*n* = 82), followed by *cps* 2 with 23.9% (*n* = 51). In descending frequency, *cps* 1/2, 4, 7, and 1 followed. While eleven lymph node isolates (5.2%) were non-typable, seven isolates (3.3%) could not be distinguished into *cps* type 2 or 1/2. The proportion of *cps* types in *S. suis* isolates from different lymph nodes showed the same distribution as in all tested sites (Appendix A). The percentage distribution of *cps* types in the different lymph nodes is shown in Figure 3.

## 4. Discussion

In this study, piglets sent for necropsy to IVD were investigated. This is generally only conducted in the case of severe herd problems. In the current study, herds with infrequent cases of arthritis and central nervous disorders are most likely under-represented. This bias most likely has an influence on the bacteriological results. As an example, *S. suis* and *T. pyogenes* were detected in 50.6% and 2.6% of joint swabs of arthritis lesions in weaning piglets, respectively. The comparable low rate of detection of *T. pyogenes* might well be related to the fact that *T. pyogenes* is less likely involved in severe herd problems [1]. Furthermore, the high number of weaning piglets examined may be because either they are sent more frequently than other age categories, they are more severely/frequently affected, or the diseases affect a larger number of animals more quickly, or they are smaller and therefore easier to send than fattening pigs. Importantly, the results of the current study clearly show the actual importance of *S. suis* in herd problems with arthritis and central nervous disorders in Germany.

### 4.1. Arthritis

In just over one-third of the joint swabs, no pathogen could be detected by bacterial culture, although pathological–anatomical arthritis was visible, and pathohistological synovitis was diagnosed in all animals. Hill et al., (1996) pathohistologically diagnosed arthritis in 165 of 175 animals studied [28]. In 31% of the animals (*n* = 51), no bacterial pathogen was detectable. This confirms that in other studies, despite pathohistological evidence of joint lesions, the aetiology could not be clarified in all cases using bacterial culture. Possible reasons for this could be that the causative pathogen had already been eliminated by the immune system at the time the lesion was sampled or it was a pathogen that was not detected by the detection methods used or it was a non-infectious arthritis.

In suckling piglets, *S. suis* was the most frequently detected pathogen, but other pathogens, such as *E. coli* and *S. hyicus*, occurred almost as frequently. In older studies, haemolytic streptococci were the causative pathogens in 63% of cases, but *T. pyogenes*, *Staphylococcus* spp., and *E. coli* were also frequent causes [29,30]. In contrast, *Erysipelothrix rhusiopathiae* (*E. rhusiopathiae*) was rarely encountered in suckling piglets with polyarthritis [29]. In this study, *T. pyogenes* was not detectable in suckling piglets, and *E. rhusiopathiae* was not isolated at all. In contrast, *S. suis* seems to have (re)gained importance and is currently the most important bacterial cause of arthritis in suckling piglets in Northern Germany.

In weaning piglets, *S. suis* was detected in more than half of the joint swabs examined culturally or by PCR. Less frequently, *M. hyorhinis*, *G. parasuis*, *T. pyogenes*, and *S. dysgalactiae* were isolated from the joint swabs of weaning piglets. According to Madson et al., (2019), the most common organisms of herd significance will vary with age but include *S. suis*, *G. parasuis*, *Mycoplasma* spp., *E. rhusiopathiae*, and *A. suis* [1]. The large proportion of arthritis due to *S. suis* in weaning piglets could be an immunological gap since maternal antibodies against *S. suis* decrease after four weeks [31], whereas, according to Baums et al., (2010), the preparturient vaccination of sows with autogenous *S. suis* bacterin might protect piglets up to the sixth week of life against disease by the homologous strain [32]. However, it may depend on the autogenous vaccine produced since field studies with other products showed that maternal antibodies induced after sow vaccination are not present during the nursery period [33,34].

While *S. suis* was the most important bacterial agent recovered in arthritis cases in suckling and weaning piglets, *M. hyosynoviae* was mostly found in fattening pigs. While the character of the inflammation was similar in most cases, the age of the host seems to have a significant influence on susceptibility to different arthritic pathogens. The results indicate a particular susceptibility of weaning piglets to arthritis caused by *S. suis* infections (age predisposition).

### 4.2. Meningitis

This study also aimed at identifying the aetiologies of central nervous diseases and clarifying the extent to which *S. suis* is the responsible pathogen. Bacterial pathogens could be detected in almost two-thirds of the meningeal swabs, with *S. suis* being found in almost 85% of the cases. Thus, *S. suis* was by far the most important pathogen in the pathogen-positive meningeal swabs of the pigs examined. A pathogen other than *S. suis* was detected in just under 10% of the meningeal swabs. The second most common pathogen was *G. parasuis*, followed by *T. pyogenes*.

The pathohistological diagnosis of meningoencephalitis or leptomeningitis was found in two-thirds of the examined CNS samples, and it was in most fibrinopurulents. Fibrinous inflammation or inflammation dominated by neutrophils suggests a bacterial genesis, whereas lymphohistiocytic and plasma cellular infiltrates may indicate viral or protozoan pathogens [35]. In *S. suis* infections, fibrinopurulent or lympholeucocytic infiltrates are present in the meninges [35,36]. In the presence of other pathohistological changes, however, other aetiologies appear more likely. Cerebrospinal angiopathy was found in more than 20.0% of the samples examined. Cerebrospinal angiopathy in porcine enterotoxaemia (oedema disease) leads to fibrinoid vessel wall necrosis with permeability disorders and secondary malacia in the CNS [35]. The samples were from weaning piglets and some fattening pigs. Oedema disease often occurs within two weeks after weaning [2]. However, it can also occur after the pigs have been introduced into the fattening system. Four pigs exhibited eosinophilic meningoencephalitis. The perivascular infiltration of eosinophilic granulocytes and cerebral oedema as well as the laminar necrosis of the middle cortical neuronal layer can occur in water withdrawal encephalopathy [35]. Accordingly, the observed changes were very likely consequences of saline poisoning or water deprivation. No sample showed signs of possible differential diagnoses, such as viral infections (e.g., Aujeszky’s disease, African and European swine fever) or meninxfibrosis [2,35].

The histological examination confirmed the suspicion of purulent meningitis in almost all cases, and it also provided evidence for the aetiology of central nervous disorders, which initially remained unclear after the macroscopic examination.

Although *S. suis* was detected in leptomeningitis and meningoencephalitis in all age classes, by far both the diseases and *S. suis* occurred most frequently in weaning piglets. The results also indicate a particular susceptibility of weaning piglets to the manifestation of inflammation by *S. suis* in the CNS.

### 4.3. S. suis Isolates

In the present study, the amplification of *gdh* was previously used as a species-specific method, and the 16S rRNA was used for species diagnosis [21,37]. However, it has been described that the detection of *gdh* is not necessarily highly specific for *S. suis* [37,38,39]. In the MP PCR *recN* was included, which is presently considered the specific marker for *S. suis* [40]. Therefore, it can be excluded that the collection contains *S. parasuis*- and *S. suis*-like isolates that are *gdh* positive.

All *S. suis* isolated were analysed for the presence of the genes *epf*, *mrp*, and *sly* using the modified MP PCR according to Silva et al., (2006) [21]. In the present study, *cps* type 2 occurred most frequently in combination with all three virulence marker genes (*epf*+ *mrp*+ *sly*+) and second most frequently with only *mrp*. Similarly, observations were previously reported for German *S. suis* field isolates [5,21]. Both *mrp* and *sly* could be detected in most *cps* types examined. This is consistent with observations described in the literature [8,41,42]. The gene *epf* was mainly observed within *cps* 1, 2, and 14. However, *epf* was also amplified in *cps* types 1/2, 8, and 9 as well as some non-typable *S. suis* isolates. The gene *epf* has so far only been found in serotype 15 in addition to serotypes 1, 1/2, 2, and 14 [8]. So, the detection of *cps* 9 with *epf* contrasts with previous results [5,8,21,43,44]. In domestic pigs, a significant proportion of *S. suis* meningitis and septicaemia in Central Europe is due to *cps* 2 *mrp*+ *sly*+ *epf*+ and *cps* 9 *mrp** strains [45,46]. Data from previous studies indicate that EF-negative serotype 2 and 1/2 strains may be less virulent or nonvirulent in pigs compared with EF-positive serotype 2 strains [8,47]. However, the detection of the gene in the PCR does not necessarily mean that the corresponding protein is expressed. Whether the presence of *epf* in *cps* 9 isolates causes an increased virulence potential remains unclear. Nevertheless, this discovery shows that the population of *S. suis* and the occurrence of virulence-associated genes are constantly changing.

*Cps* types 2 and 9 were detectable in 243 of the 395 *S. suis* isolates and together were the most frequently detected serotypes. Comparing *S. suis* strains from between 1996 and 2004 and between 2015 and 2016 in Germany, Pruefer et al., (2019) reported that *cps* types 2 or 1/2 and *cps* type 9 were classified as equally important as invasive infectious agents for pigs in Germany [5]. While classical serotyping was of prior importance in diagnostics until a few years ago, molecular biological methods are becoming increasingly important in practice. Several published PCR assays were designed for detecting [21] many or all important serotypes of *S*. *suis* [24,48,49]. It has been criticized that PCR assays used for epidemiological studies cannot differentiate between *cps* 1 and *cps* 14, as well as between *cps* 2 and *cps* 1/2 [50,51]. Therefore, the specific prevalence of each of these four serotypes is currently not known in Germany because a recently published study is based on MP PCR [5]. In the current study, a selected number of *cps* 1 or 14 isolates were distinguished by sequencing, while *cps* 2 or 1/2 isolates were distinguished by MAMA, with the result that *cps* 1 was noticeably more frequently detected than *cps* 14. In addition, the proportion of *cps* 2 was about three times larger than the proportion of *cps* 1/2, which is comparable to the results of Allgaier et al., (2001) [45]. Overall, it could be shown that *cps* 9 (*n* = 159) occurred one and a half times as often as *cps* 2 (*n* = 107); therefore, *cps* 9 is currently the most common *cps* type in Germany. This is consistent with the situation in other European countries, such as the Netherlands, Spain, and Switzerland [52,53,54,55,56].

### 4.4. Occurrence of S. suis by Animal and Age Category

*S. suis* was detected in all animal categories. Indeed, *S. suis* is a pathogen that may affect pigs of all ages, although it is rarely observed in fattening animals [3]. Most clinical disease is observed during the nursery period [1,7]. This could not be confirmed by the present study. Although far more weaning piglets than suckling piglets and fattening pigs were examined, *S. suis* was only slightly more detectable in weaning piglets than in the other age categories. There was only a trend that with advancing age *S. suis* could be detected to a decreasing extent (Table 4). Even though *S. suis* can occur in different age categories, many epidemiological studies do not report the age data of the affected pigs [5,21]. A previous study suggested differences between *S. suis* serotypes regarding the age of affected piglets [8]. We compared the occurrence of *cps* types among invasive German isolates of suckling piglets, weaning piglets, and fattening pigs with musculoskeletal and central nervous disorders. While *cps* 1 and *cps* 7 dominated in the suckling piglets, *cps* 9 and *cps* 2 were most frequently detected in the weaning piglets (compare [27]).

### 4.5. Occurrence of S. suis in Lymph Nodes

In this study, eight different lymph nodes from 201 animals with musculoskeletal and central nervous disorders were systematically examined. As shown in Table 5 and Appendix A, various lymph nodes might be infected with *S. suis.* In piglets with a disseminated infection, the same *S. suis* genotype was detectable in multiple lymph nodes. These results suggest that the host–pathogen interaction of *S. suis* in lymph nodes is an important part of the pathogenesis and that immune evasion mechanisms of *S. suis* allow this pathogen to survive in lymph nodes for longer time periods. In 52 animals, more than one lymph node was positive. Lymph nodes might have contained *S. suis*-positive lymphocytes. *S. suis* has been shown to adhere to lymphocytes [57]. Due to the special anatomy of porcine lymph nodes, an exchange of lymphocytes between the blood and the lymph nodes is possible via the high-endothelial venules [58]. Many *S. suis*-positive lymph nodes might indicate that *S. suis* targets these tissues during systemic infection.

This study was originally designed to identify the putative portal of entry through bacterial screenings of lymph nodes. An infected lymph node was considered to indicate that the respective tributary region was infected first [35]. The same was assumed in studies on gastrointestinal translocation because mesenteric lymph nodes were also examined [18,19,20]. Although, in 25 animals, only one lymph node was positive, in 21 animals, one lymph node and at least one other internal organ was *S. suis* positive. If a lymph node and other internal organs were positive, no statement can be made as to the portal of entry of *S. suis*.

In two animals with only one *S. suis* positive lymph node, it was the mesenteric one, suggesting that gastrointestinal translocation may occur in naturally infected pigs. The intestines of weaning piglets are rapidly colonized by *S. suis*, and the relative intestinal concentration of *S. suis* has been reported to increase [59] or remain steady [60] after weaning. So, the gastrointestinal tract cannot be excluded as a secondary site of infection in piglets [61]. However, in the current study, there are no indications that the gastrointestinal route of infection is of great relevance.

Furthermore, *cps* 2 *mrp*+ *sly*+ *epf*+ and *cps* 9 *mrp*+ *sly*+ *epf*− were detectable in almost two-thirds of the animals in more than three sites. At the same time, two or more lymph nodes and different other internal organs were positive in 15 and 18 animals, respectively. This shows that *S. suis* infections with these two genotypes lead increasingly to systemic infections compared to many other genotypes. This made it difficult to say via which route the pathogen had entered the animal body. However, it underlines that these two genotypes are the most important in Europe.

*S. suis* isolates were detectable in the lymph nodes of the gastrointestinal tract, which entered the pig via the gastric and jejunal mucosa. Thus, the present study confirms that under field conditions, the entry of *S. suis* into the animal body via the gastrointestinal tract is possible. At the same time, the *S. suis* isolates detected appear to be clinically less relevant, as they were mainly not *cps* 2 *mrp*+ *sly*+ *epf*+ and *cps* 9 *mrp*+ *sly*+ *epf*−.

## 5. Conclusions

Our results demonstrate that *S. suis* is currently the major causative agent of arthritis and meningitis in weaning piglets in Germany, at least in herds which experience substantial herd problems associated with these pathologies. There are age-specific differences concerning the frequency of the detectable *cps* types. A specific *S. suis* genotype is often detected in multiple lymph nodes in a single animal. Accordingly, the examination of the lymph nodes does not help clarify the portal of entry for *S. suis* in these cases. The results suggest that gastrointestinal translocation might occur in naturally infected pigs, but there are no indications this route of infection is of great relevance.

## Figures and Tables

**Figure 1 vetsci-11-00017-f001:**
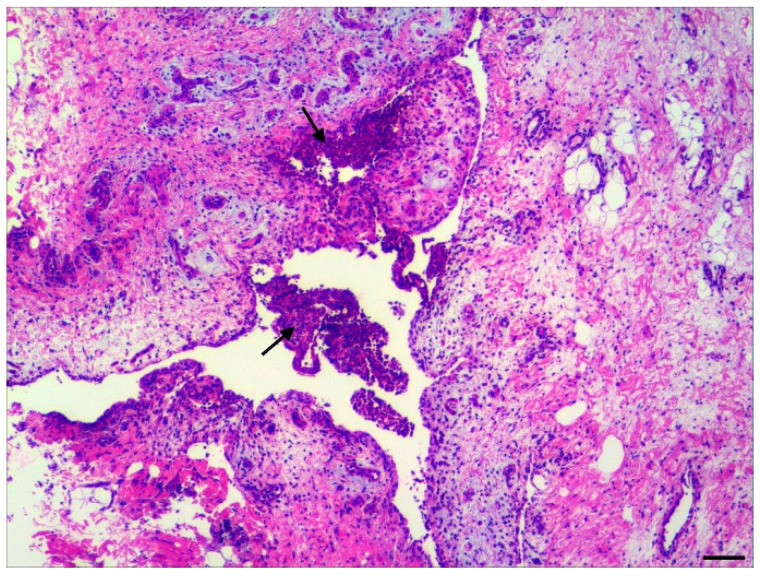
Suckling piglet. Joint capsule. Fibrinopurulent arthritis. The joint capsule is expanded by marked inflammation. Luminal exudate consisting of neutrophilic granulocytes and fibrin (arrows). Haematoxylin–eosin stain. 4×. Bar length = 78 µm.

**Figure 2 vetsci-11-00017-f002:**
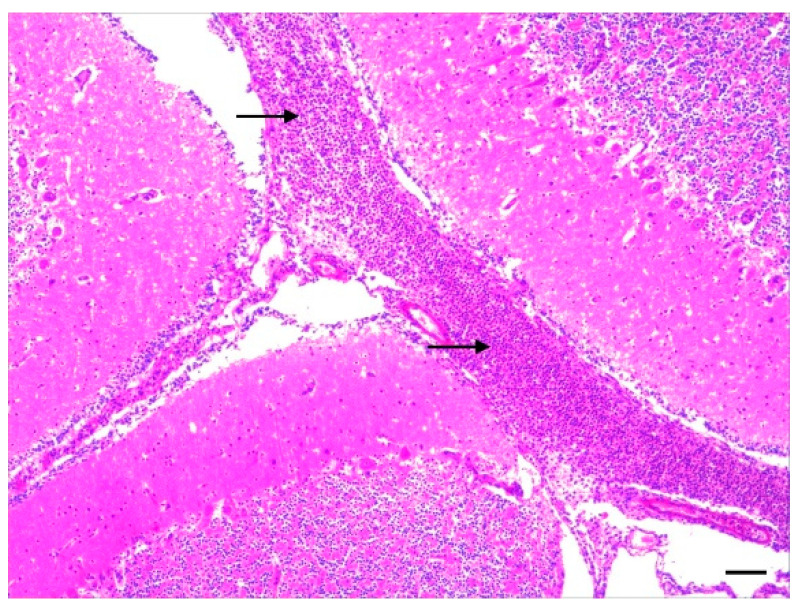
Suckling piglet. Cerebellum. Fibrinopurulent leptomeningitis. Leptomeningeal inflammatory infiltrates consisting of neutrophilic granulocytes, fibrin, and necrotic debris (arrows). Haematoxylin–eosin stain. 4×. Bar length = 135 µm.

**Figure 3 vetsci-11-00017-f003:**
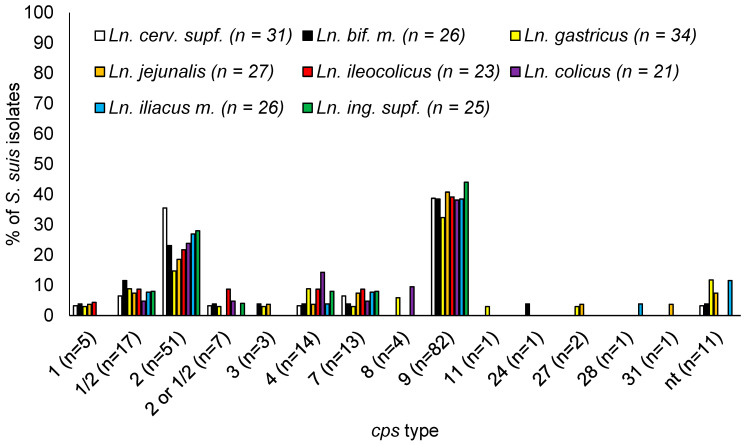
Proportions of *cps* types between *S. suis* isolates from different lymph nodes (per lymph node and per herd each different genotype only once) based on the detection of serotype-specific genes of the *cps* locus. Non-typable *S. suis* isolates (nt) may be unencapsulated or belong to a new *cps* locus (NCLs). Total of 213 *S. suis* isolates.

**Table 1 vetsci-11-00017-t001:** Target genes and amplicons of multiplex PCR for the identification of *S. suis* and for the detection of *cps* types 1 or 14, 2 or 1/2, 7, and 9 as well as the virulence markers EF, MRP, and SLY.

Target Gen	GenBank Accession Number	Amplicon [bp]	Reference
*gdh*	CP000408	954	IVD GmbH (Validation 2012)
*epf*	X71881	708	IVD GmbH (Validation 2012)
*cps* 1 or 14	AF155804	555	IVD GmbH (Validation 2012)
*cps* 2 or 1/2	AF118389	443	based on [21]
*cps* 7	AF164515	379	[21]
*cps* 9	AF155805	303	[21]
*sly*	Z36907	248	[21]
*mrp*	X64450	188	[21]
*srtD*	AB066354	139	based on [22]

*gdh*: glutamate dehydrogenase gene; *epf*: extracellular factor (EF) gene; *cps*: capsular genes; *sly*: suilysin (SLY) gene; *mrp*: muramidase-released protein (MRP) gene; *srtD*: sortase D gene.

**Table 2 vetsci-11-00017-t002:** Pathogen detection in 120 joint swabs by cultural examination with age category.

Result of Bacteriological Culture of Joint Swab (%)	Age Category (Number of Examined Joint Swabs)			
	Suckling Piglets (*n* = 25 *)	Weaning Piglets (*n* = 77)	Fattening Pigs (*n* = 18)	Total (*n* = 120)
** *Escherichia coli* **	3 (12.0)	0	0	3 (2.5)
***Escherichia coli* and *Staphylococcus hyicus***	1 (4.0)	0	0	1 (0.8)
***Escherichia coli* and *Streptococcus suis***	1 (4.0)	0	0	1 (0.8)
** *Glaesserella parasuis* **	0	3 (3.9)	1 (6.0)	4 (3.3)
***Helcococcus* spp.**	1 (4.0)	0	0	1 (0.8)
** *Mycoplasma hyorhinis* **	0	5 (6.5)	0	5 (4.2)
** *Mycoplasma hyosynoviae* **	0	0	1 (6.0)	1 (0.8)
***Proteus* spp.**	0	1 (1.3)	0	1 (0.8)
***Salmonella* spp.**	0	1 (1.3)	0	1 (0.8)
** *Staphylococcus aureus* **	3 (12.0)	1 (1.3)	0	4 (3.3)
** *Staphylococcus hyicus* **	2 (8.0)	0	0	2 (1.6)
** *Streptococcus dysgalactiae* **	2 (8.0)	2 (2.6)	0	4 (3.3)
** *Streptococcus suis* **	5 (20.0)	39 (50.6)	1 (6.0)	45 (37.5)
** *Trueperella pyogenes* **	0	2 (2.6)	1 (6.0)	3 (2.5)
**negative**	7 (28.0)	23 (29.9)	14 (78.0)	44 (36.7)
**total**	25 (100.0)	77 (100.0)	18 (100.0)	120 (100.0)

* Twenty-five joint swabs from 24 suckling piglets. spp.: subspecies.

**Table 3 vetsci-11-00017-t003:** Pathogen detection in 133 meningeal swabs by cultural examination according to pathohistological diagnosis and age catergory.

Diagnosis of Pathohistological Examination	Result of Bacteriological Culture of Meningeal Swab	Age Category (Number of Examined Meningeal Swabs)			
		Suckling Piglets (*n* = 7)	Weaning Piglets (*n* = 109)	Fattening Pigs (*n* = 17)	Total (*n* = 133)
**Leptomeningitis**	** *Eschericia coli* **	1	0	0	1
	** *Glaesserella parasuis* **	0	3	0	3
	** *Streptococcus suis (S. suis)* **	2	44 *	0	46
	** *Staphylococcus aureus (S. aureus)* **	0	2	0	2
	** *Trueperella pyogenes* **	2	1	0	3
	**negative**	1	4	1	6
	**total**	**6**	**54**	**1**	**61**
**Meningoencephalitis**	** *S. suis* **	0	13	2	15
	** *Streptococcus dysgalactiae* **	0	0	1	1
	**negative**	1	6	4	11
	**total**	**1**	**19**	**7**	**27**
**Cerebrospinal angiopathy**	** *S. suis* **	0	1	0	1
	***S. suis* and *S. aureus***	0	1	0	1
	** *Actinobacillus minor* **	0	1	0	1
	***Enterococcus* spp.**	0	1	0	1
	**negative**	0	19	5	24
	**total**	**0**	**23**	**5**	**28**
**Hyperaemia and without lesions**	** *S. suis* **	0	1	0	1
	**negative**	0	12	4	16
	**total**	**0**	**13**	**4**	**17**

* One weaning piglet with leptomeningitis and cerebrospinal angiopathy. spp.: subspecies. Bold numbers highlight the total number of animals with the respective diagnoses and age category.

**Table 4 vetsci-11-00017-t004:** Frequency of *S. suis* detections by age category.

		Suckling Piglet	Weaning Piglet	Fattening Pig	Total
**pigs examined**	*n*	27	144	30	201
	%	13.4	71.6	14.9	100.0
**pigs with cultural detection of *S. suis***	*n*	16	93	15	124
	%	12.9	75.0	12.1	100.0
**A**	%	59.3	64.6	50.0	61.7

A: Proportion of animals with bacterial culture detection of *S. suis* in all examined animals of the respective age category. n: number. %: percentage.

**Table 5 vetsci-11-00017-t005:** Detection of *S. suis* isolates in lymph nodes depending on age category.

Lymph Node (Number of Examined Lymph Nodes)	No. of *S. suis* Detections per Age Category			Total No. of *S. suis* Detections	
	Suckling Piglets	Weaning Piglets	Fattening Pigs	No.	(%)
*Ln. cervicalis superficialis dorsalis dexter* (*n* = 201)	3	36	2	41	20.4
*Ln. bifurcationis medius* (*n* = 201)	1	30	4	35	17.4
*Ln. gastricus* (*n* = 201)	1	32 *	4	37	18.4
*Ln. jejunalis* (*n* = 201)	2	28	1	31	15.4
*Ln. ileocolicus* (*n* = 201)	1	27	0	28	13.9
*Ln. colicus* (*n* = 162)	3	21	0	24	14.8
*Ln. iliacus medialis dexter* (*n* = 201)	1	31	2	34	16.9
*Ln. inguinalis superficialis dexter* (*n* = 201)	2	35 *	1	38	18.9
total (*n* = 1569)	14	240	14	268	17.8

* Within one lymph node two different *S. suis* isolates. *Ln.*: *Lymphonodus.*

## Data Availability

Data are contained within the article and Appendix A.

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
