# Peer review of "Invasive Bacterial Infections of the Musculoskeletal and Central Nervous System during Pig Rearing: Detection Frequencies of Different Pathogens and Specific Streptococcus suis Genotypes"

_vetsci, 2024, doi:10.3390/vetsci11010017_

Round 1

Reviewer 1 Report

Comments and Suggestions for Authors

1. Histopathological studies were performed in this study, but representative images were not shown in the results.

2.Different tissue samples are collected in different quantities, please explain it.

Comments on the Quality of English Language

 Please check the typos, punctuation errors, spacing and grammar tense mistakes within the manuscript. Please modify them.

Reviewer 2 Report

Comments and Suggestions for Authors

Line 98: instead of just „heart valve“, I would prefer „endocard“ and add heart valve in brackets

Line 99: pathologically-anatomically – gross pathological

Table 1: please write all abbreviations in full as a legend to the table

Line 115: S. epidermidis in stead of St. epidermidis

Line 120: please indicated the PCRs for mycoplasma differentiation in more detail

Line 135: examinated (not: exanimated)

In section 2.7. the limit for statistical significance should be mentioned for the tests performed.

In section 3.1. it should be mentioned, whether or which animals (only those showing inflammatory signs of diverse serosas?), respectively, were cachectic or exhibited other signs of systemic disease (subcutaneous bleeding etc), which would be of interest for estimating the clinical importance of the diagnosed bacteria.

Line 175: “… lesions of leptomeningitis and lesions of cerebrospinal angiopathy. Lesions of…“. These sentences should be reformulated like „lesions indicative of leptomeningitis etc“

Line 178: “… findings … were found…” reformulate like “…. signs … were found…”

Line 179 ff: “… their character was fibropurulent…” the complete text should be checked for wording to avoid germanisms, which are named the first time (G. parasuis and T. pyogenes), the generic names should be written fully

Line 203 vs 204: was it 21 or 24 piglets?

Line 223, 224: this multiplex PCR is not mentioned, at least not directly, in the M&M section

Table 4: I do not understand the difference between the two last lines “A”

Line 456:  PCR, not pCR

Line 470: mrp+ instead of mrp*

Line 471: please indicate what EF is standing for

Lines 533-537: I can only assume what this sentence should mean. Please reformulate more precisely.

Comments on the Quality of English Language

There are a lot of "germanisms" in the text, which should be corrected by a native speaker.
